# Tuning the performance of a micrometer-sized Stirling engine through reservoir engineering

Niloyendu Roy [1✉], Nathan Leroux [2], A. K. Sood[3,4] & Rajesh Ganapathy [4,5]

Colloidal heat engines are paradigmatic models to understand the conversion of heat into work in a noisy environment - a domain where biological and synthetic nano/micro machines function. While the operation of these engines across thermal baths is well-understood, how they function across baths with noise statistics that is non-Gaussian and also lacks memory, the simplest departure from the thermal case, remains unclear. Here we quantified the performance of a colloidal Stirling engine operating between an engineered memoryless non-Gaussian bath and a Gaussian one. In the quasistatic limit, the non-Gaussian engine functioned like a thermal one as predicted by theory. On increasing the operating speed, due to the nature of noise statistics, the onset of irreversibility for the non-Gaussian engine preceded its thermal counterpart and thus shifted the operating speed at which power is maximum. The performance of nano/micro machines can be tuned by altering only the nature of reservoir noise statistics.

[1] Chemistry and Physics of Materials Unit, Jawaharlal Nehru Centre for Advanced Scientific Research, Jakkur, Bangalore, India. [2] Unité Mixte de Physique CNRS/Thales, Palaiseau, France. [3] Department of Physics, Indian Institute of Science, Bangalore, India. [4] International Centre for Materials Science, Jawaharlal Nehru Centre for Advanced Scientific Research, Jakkur, Bangalore, India. [5] School of Advanced Materials (SAMat), Jawaharlal Nehru Centre for Advanced Scientific Research, Jakkur, Bangalore, India. ✉email: niloycrj@gmail.com

Experimental advances in nano/micro manipulation have made feasible the realization of mesoscale heat engines with only a single atom[1] or colloidal particle[2–6] as the working substance. Even while the functioning of these engines is strongly influenced by fluctuations in the local environment with quantities like work and efficiency becoming stochastic, when operating between thermal/Gaussian heat baths, their cycle-averaged performance mirrors their macroscopic counterparts, and standard thermodynamic relations apply[7–12]. Recently, Krishnamurthy et al.[13] experimentally realized an active stochastic heat engine by replacing the isothermal branches of a conventional Stirling cycle with isoactive ones. Here, a colloidal particle in a time-varying optical potential was periodically cycled across two bacterial reservoirs characterized by different levels of activity. Unlike in thermal baths where the displacement distribution of the colloid, $\rho(x)$, is Gaussian, in active reservoirs, it was non-Gaussian and heavy-tailed[13,14]. These rare large displacement events resulted in large work output and the efficiency of this active engine was found to surpass equilibrium engines; even those operating between thermal baths with an infinite temperature difference. Since the metabolic activity of the bacteria could not be altered rapidly, this engine was operated only in the quasistatic limit, i.e., for a cycle duration $\tau$ larger than the relaxation time of the colloid. Subsequent theoretical calculations for the $\tau \to \infty$ limit posited that a departure from equilibrium efficiencies requires noise not just with non-Gaussian statistics but also with memory, a feature typical of active baths due to the persistent motion of the particles[15]. In fact, when the bath noise is non-Gaussian and white, an effective temperature $T_{\text{eff}}$ defined through the variance of $\rho(x)$ is thought to act like a bona fide temperature[15,16] and engines operating between such baths are expected to perform like thermal ones in the quasistatic limit. Whether this similarity persists when $\tau$ is reduced and irreversibility begins to set in is not known and is worth exploring since real heat engines never operate in the quasistatic limit as here their power $P \to 0$. On the experimental front, memoryless non-Gaussian heat baths are yet to be realized and predictions even in the quasistatic limit remain untested.

Here we engineered non-Gaussian heat baths without memory ($\delta$-correlated noise) and with different kurtosis, $\kappa$, and then operated colloidal Stirling heat engines between these baths and thermal ones for different $\tau$. In the quasistatic limit, the performance of these non-Gaussian engines mirrored a classical Stirling engine operating between thermal/Gaussian baths in agreement with theoretical predictions. Strikingly, due primarily to differences in the noise statistics of the baths, the small $\tau$ behavior of these engines was quite different. On lowering $\tau$, not only did the distribution of work done per cycle, $\rho(W_{\text{cyc}})$, for the non-Gaussian engines become increasingly negatively skewed, unlike the standard Stirling case where it remained Gaussian, the onset of irreversibility for these engines was also different. Importantly, we demonstrate that even without memory, changing the nature of noise statistics of the reservoirs between which an engine operates allows tuning its performance characteristics, specifically, the $\tau$ at which the power goes through a maximum.

## Results

### Reservoir engineering by flashing optical traps
Our experimental scheme for reservoir engineering is elaborated in Fig. 1a. A polystyrene colloidal particle of radius $R = 2.5$ μm suspended in water is held in a harmonic optical potential, $U = \frac{1}{2} k_1 \langle x^2 \rangle$, created by tightly focusing a laser beam (1064 nm ALS-IR-5-SF, Azur Light Systems France) through a microscope objective (Leica Plan Apochromat 100×, N.A. 1.4, oil) that is also used for imaging the particle (see "Methods"). Here, $k_1$ is the stiffness of

this primary trap, $x$ is the displacement of the colloid from the center of the optical trap and $\langle \rangle$ denotes an average. At equilibrium, the trap stiffness can be determined through the equipartition relation $\frac{1}{2} k_1 \langle x^2 \rangle = \frac{1}{2} k_{\text{B}} T$ where $k_{\text{B}}$ is the Boltzmann constant and $T$ is the bath temperature, which in our experiments is fixed at 300 K. As a first step, we attempted to engineer a reservoir that mimicked a thermal bath, i.e., with Gaussian noise statistics, but with the desired $T_{\text{eff}}$. To this end, we imposed an additional noise on the colloidal particle along one spatial dimension, here the x-axis (Fig. 1a), from a second optical trap of fixed intensity but with a time-dependent center that was flashed at a distance $\delta a(t)$ away from the primary one (Fig. 1b). This was made possible by using a second laser (Excelsior 1064 nm, Spectra Physics USA) coupled to the microscope through a spatial light modulator (SLM). The refresh rate of the SLM set the speed at which the secondary traps could be flashed and to ensure that our findings were not sensitive to the SLM's refresh rate, experiments were carried out with both a low-speed SLM (Boulder Nonlinear Systems USA) with a flashing frequency of 34 Hz and a high-speed SLM (Meadowlark Optics USA) with a flashing frequency of 135 Hz (see "Methods"). Earlier reservoir engineering studies wherein the colloidal particle experienced only the potential from the flashing trap found that when $\delta a$ was drawn from a Gaussian distribution, the particle indeed behaved like one in a thermal bath but at a $T_{\text{eff}} > T$ and furthermore, when $\delta a(t) < R$, the trap stiffness also remained unaltered[17,18]. Here, we adhered to the same protocol and further ensured that the peak of the $\delta a$ distribution coincided with the center of the primary trap. Thus, the effective trap stiffness in our experiments $k = k_1 + k_2$, where $k_2$ is the stiffness of the flashing trap. Like in a thermal bath, $\rho(x)$ of the trapped colloidal particle was a Gaussian (solid circles in Fig. 1d), and its power spectral density (PSD) a Lorentzian, allowing us to determine $k_2$ and hence $T_{\text{eff}}$[18] (Supplementary Fig. 1 and Supplementary Note 1). For the $\delta a(t)$ profile shown in Fig. 1b, the particle experienced a $T_{\text{eff}} = 1331$ K.

Engineering a memoryless non-Gaussian reservoir involved only a small tweak to the manner in which the external noise was imposed on the colloidal particle. The instantaneous $\delta a$ was now drawn randomly from a distribution with zero mean and skew, as before, but with a high $\kappa$ (see "Methods" and Supplementary Fig. 2). Such a distribution has a narrow central region with heavy tails. The flashing optical trap is thus mostly coincident with the primary trap, thereby confining the particle strongly, and is occasionally positioned a large distance away from the center leading to a large excursion by the particle (Fig. 1c and Supplementary Movie). The overall noise experienced by the particle is $\delta$-correlated as the thermal and imposed noise are individually $\delta$-correlated. Under the influence of such a noise, the corresponding $\rho(x)$ of the colloidal particle was also non-Gaussian. The hollow squares in Fig. 1d show $\rho(x)$ ($\kappa = 27$) for a flashing frequency of 34 Hz and the hollow triangles in Supplementary Fig. 4a show $\rho(x)$ ($\kappa = 10$) for a flashing frequency of 135 Hz. The PSD of the trapped particle, for both the flashing frequencies, could be fit to a Lorentzian with the fit showing better agreement with the data over a broader dynamic range for the higher flashing frequency. This suggests that the overall noise experienced by the particle is indeed uncorrelated and additive. Since all other experimental parameters are held fixed, the roll-off frequency of the PSD was also same as that of the Gaussian case (Supplementary Fig. 3 and Supplementary Note 1). For an appropriate choice of the variance and kurtosis of the $\delta a$ distribution, we could engineer the $T_{\text{eff}}$ of the non-Gaussian bath, again defined through the variance of $\rho(x)$, to be nearly identical to that in a Gaussian bath (Fig. 1d).

### Performing a Stirling cycle between engineered reservoirs
Armed with the capability to engineer reservoirs, we first built a

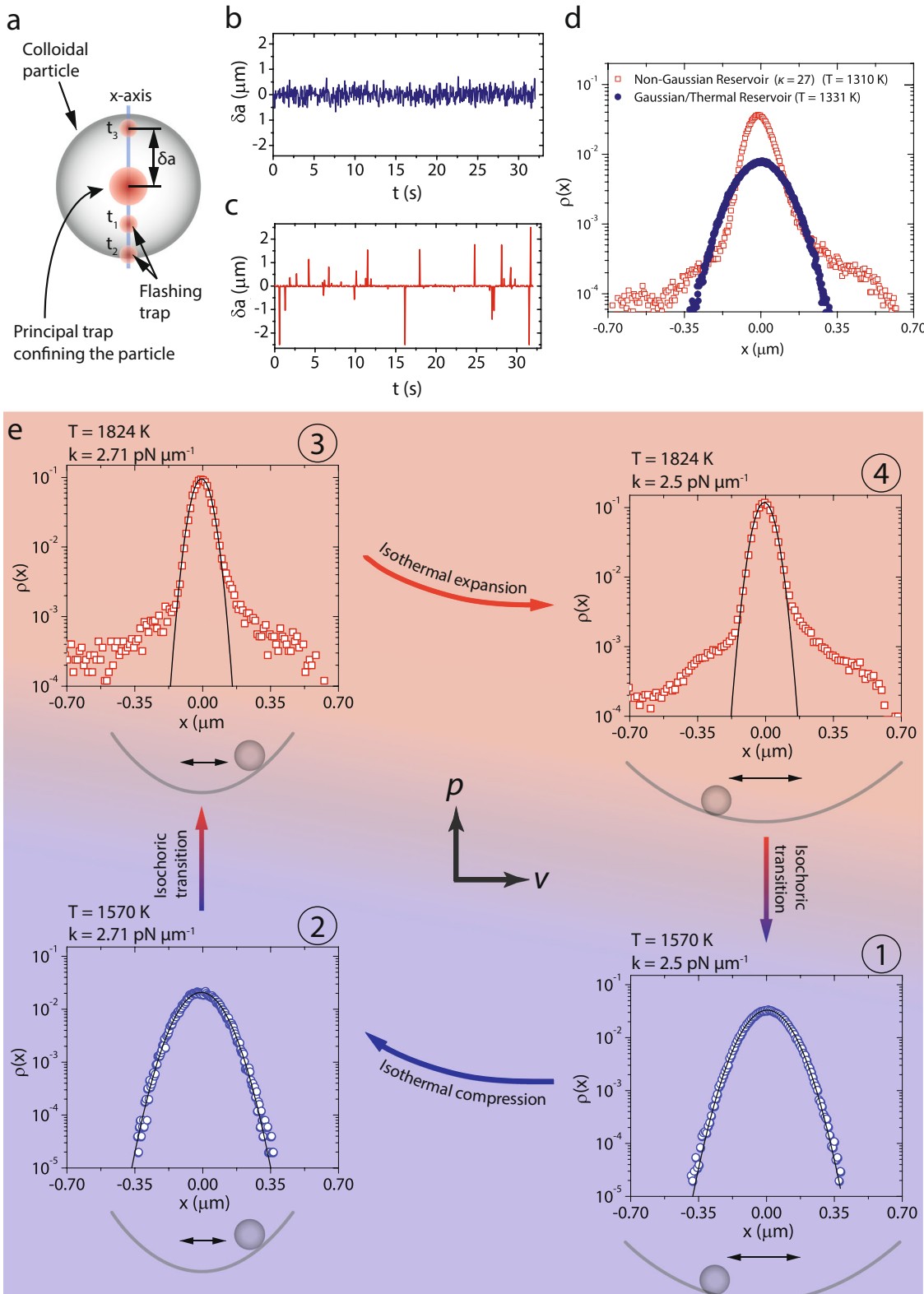

**Fig. 1 Experimental realization of a non-Gaussian Stirling heat engine. a** The big red spot represents the primary optical trap and the small red spots represent the secondary flashing optical trap at different time instances $t_1$–$t_3$. **b, c** The distance $\delta a(t)$ from the primary trap at which the secondary trap was flashed as a function of $t$ for engineering a Gaussian and a non-Gaussian reservoir, respectively. **d** The probability distribution of particle displacements, $\rho(x)$, for the engineered Gaussian/thermal (solid blue circles) and the non-Gaussian reservoir with $\kappa = 27$ (red hollow squares) for a nearly identical $T_{\rm eff}$. **e** A quintessential Stirling cycle between a hot non-Gaussian ($\kappa = 20$) bath at $T_{\rm eff}^{\rm H} = 1824$ K and a cold Gaussian reservoir with $T_{\rm eff}^{\rm C} = 1570$ K. The trap stiffness $k$ is varied linearly in the expansion/compression steps. Having a fixed primary trap and a second flashing optical trap, as opposed to just the latter, prevented the trapped particle from escaping the trap and allowed for long experiments. $\rho(x)$ of the particle measured at the four-state points (at equilibrium) labeled ①–④ is also shown. The black lines are Gaussian fits.

colloidal Stirling engine operating between a hot non-Gaussian ($\kappa = 20$) and a cold Gaussian bath held at temperatures $T_{\text{eff}}^{\text{H}} = 1824\,\text{K}$ and $T_{\text{eff}}^{\text{C}} = 1570\,\text{K}$, respectively. We then compared the performance of this engine with yet another non-Gaussian engine operating between a hot reservoir ($T_{\text{eff}}^{\text{H}} = 1500$ K, $\kappa = 10$) and a cold Gaussian reservoir ($T_{\text{eff}}^{\text{C}} = 1140$ K) (see Supplementary Fig. 4 and Supplementary Note 4) and also a standard Stirling engine operating between engineered Gaussian baths with $T_{\text{eff}}^{\text{H}} = 1378\,\text{K}$ and $T_{\text{eff}}^{\text{C}} = 1238\,\text{K}$ (see Supplementary Note 2). The Stirling cycle we executed with the trapped colloid (Fig. 1e), like in the previous studies[2,13,19], comprised of an isothermal compression (path ①–②) and expansion step (path ③–④) linked by two isochoric transitions (paths ②–③ and ④–①). In the isothermal compression (expansion) steps, $k$ was increased (decreased) linearly by changing $k_1$ alone (see Fig. 1e and Supplementary Note 4). The isochroric transitions were nearly instantaneous and occurred on millisecond time scales. We exploited the ability to rapidly alter $T_{\text{eff}}$ and also the nature of noise statistics through the SLM to explore engine performance over a range of $\tau$ which spanned from 2 to 32 s (see "Methods").

**Elucidating the origins of irreversibility in the non-Gaussian Stirling engine.** The framework of stochastic thermodynamics provides a prescription for calculating thermodynamic quantities like the work, power, and efficiency of mesoscopic machines[7,8,10,19]. The work done per cycle, $W_{\text{cyc}}$, by the particle due to a modulation in the stiffness of the trap is just the change in potential energy and is given by $W_{\text{cyc}} = \int_{t_i}^{t_i+\tau} \frac{\partial U}{\partial k} \circ dk \equiv \frac{1}{2} \int_{t_i}^{t_i+\tau} x^2 \circ dk$. Here, the $\circ$ signifies that the product is taken in the Stratonovich sense and $t_i$ is the starting time of $i$th cycle. Owing to its stochastic nature, $W_{\text{cyc}}$ of the engine fluctuates from cycle-to-cycle and we quantified the nature of these fluctuations through the probability distribution function $\rho(W_{\text{cyc}})$. Figure 2a, b shows $\rho(W_{\text{cyc}})$ at different $\tau$ for the thermal and non-Gaussian ($\kappa = 20$) Stirling cycles, respectively (see Supplementary Fig. 4b for a non-Gaussian engine with $\kappa = 10$ for the hot reservoir). Focusing on the large cycle duration ($\tau = 18.8$ s) first, we observed that $\rho(W_{\text{cyc}})$ is a Gaussian for the thermal and also for the non-Gaussian cycles (circles in Fig. 2a, b and see Supplementary Fig. 4b). The experimentally calculated average work done per cycle, $\langle W_{\text{cyc}} \rangle$, is negative indicating that the engine extracts heat from the bath to perform work on the surroundings. Further, $\tau = 32$ s corresponds to the quasistatic limit for all the engines since the value of $\langle W_{\text{cyc}} \rangle$ is in excellent agreement with the theoretically calculated quasistatic Stirling work output, $W_{\infty} = k_B(T_{\text{eff}}^{\text{C}} - T_{\text{eff}}^{\text{H}}) \ln\sqrt{\frac{k_{\max}}{k_{\min}}}$ (short solid horizontal lines in Fig. 2c and Supplementary Fig. 4c).

On lowering $\tau$, $\rho(W_{\text{cyc}})$ for the thermal Stirling engine remained Gaussian (Fig. 2a) and $\langle W_{\text{cyc}}(\tau) \rangle \approx \langle W_{\text{cyc}}(\tau = 32\,\text{s}) \rangle$ (hollow circles Fig. 2c). As expected of such a distribution, $\langle W_{\text{cyc}} \rangle$ was the same as the most probable work $W^*$—the value of $W_{\text{cyc}}$ where $\rho(W_{\text{cyc}})$ is a maximum (solid circles Fig. 2c). For both the non-Gaussian engines on the other hand, on reducing $\tau$, $\rho(W_{\text{cyc}})$ became increasingly negatively skewed (Fig. 2b and Supplementary Fig. 4b) and $W^*(\tau)$ also became increasingly positive (solid squares Fig. 2c and solid triangles in Supplementary Fig. 4c). $\langle W_{\text{cyc}}(\tau) \rangle$ however, was only marginally smaller than $\langle W_{\text{cyc}}(\tau = 32\,\text{s}) \rangle$ (hollow squares Fig. 2c and hollow triangles in Supplementary Fig. 4c). We note that the work done by a thermal Stirling engine at a finite $\tau$ is given by the relation[2,19]

$$W(\tau) = W_{\infty} + W_{\text{diss}} \equiv W_{\infty} + \frac{\Sigma}{\tau} \qquad (1)$$

where $W_{\text{diss}}$ is the dissipative work which accounts for the particle's inability to fully explore the available phase space when $k$ is rapidly lowered during the hot isotherm and $\Sigma$ is a constant

also called the irreversibility parameter. Since $W_{\text{diss}}$ is a positive quantity as per definition, at small enough $\tau$, the overall work done itself can be positive indicating the stalling of the engine. Clearly, there is no buildup of irreversibility for the thermal engine as $\tau$ is lowered since $\langle W_{\text{cyc}}(\tau) \rangle \equiv W^*(\tau) \approx W_{\infty}$, while for the non-Gaussian one, there is, even if only in the most-probable sense ($\langle W_{\text{cyc}}(\tau) \rangle \approx W_{\infty} < W^*(\tau)$), and the engine stalls for $\tau \lesssim 10$ s. We also found excellent agreement between equation (1) and our data allowing us to determine $\Sigma = 0.11\,k_B T_{\text{eff}}^{\text{C}}$ (red solid line in Fig. 2c). Furthermore, the work output of the non-Gaussian engine with $\kappa = 10$ for the hot reservoir also showed a similar behavior (Supplementary Fig. 4c and Supplementary Note 4) with irreversibility building up at comparatively smaller $\tau$, resulting in positive $W^*$ (stalling) for $\tau \lesssim 6$ s.

The observed behavior of the non-Gaussian engines can be easily rationalized by analyzing the relaxation of the particle in the hot isotherm at the level of an individual cycle. For the particle to fully sample the statistical properties of the non-Gaussian hot reservoir, it should also experience the occasional large kicks that displace it far from the center and not just the ones that predominantly keep it confined close to it. As $\tau$ is lowered, in most cycles, the probability that the particle encounters a large kick in the isothermal expansion step also becomes increasingly small. Due to the incomplete exploration of the available phase volume in these cycles, less useful work is performed and $W^*(\tau)$ lifts off with decreasing $\tau$. In a few cycles, where these large kicks are present, anomalously large work is done by the engine and this results in $\rho(W_{\text{cyc}})$ being negatively skewed. When an adequate number of cycles, which has to be increased when $\tau$ is lowered, has been performed, all features of the noise are sampled and the engine operates like one in the quasistatic limit in an average sense with $\langle W_{\text{cyc}}(\tau) \rangle \rightarrow W_{\infty}$ (Fig. 2c). More interestingly, a comparison of non-Gaussian engines with $\kappa = 20$ (Fig. 2c) and $\kappa = 10$ (Supplementary Fig. 4c) for the hot reservoir, respectively, allows us to infer that this irreversibility due to the unavailability of large kicks in the isothermal expansion step is also dependent upon the extent of non-Gaussianity (Supplementary Note 4). This inference can be further strengthened by quantifying the equilibration of the particle over a fixed, but limited, number of cycles for all $\tau$. In Fig. 2d, we show $\frac{k\langle x^2 \rangle}{k_B T_{\text{eff}}^{\text{H}}}$ calculated over a small window in the middle of the hot isotherm and averaged over $N = 50$ cycles for the thermal (squares) and the non-Gaussian ($\kappa = 20$) engine (circles). Despite $N$ being small, $\frac{k\langle x^2 \rangle}{k_B T_{\text{eff}}^{\text{H}}}$ is close to 1 at all $\tau$ for the thermal engine implying that it is operating in the quasistatic limit, while for the non-Gaussian engine ($\kappa = 20$) this is the case only at large $\tau$ with a clear violation of quasistaticity setting in for $\tau \lesssim 10$ s. Evidently, for a non-Gaussian engine $W^*(\tau)$, and not $\langle W_{\text{cyc}}(\tau) \rangle$, is a more precise metric for performance. To further strengthen this claim, we plot $\frac{k\langle x^2 \rangle}{k_B T_{\text{eff}}^{\text{H}}}$ at the middle of the hot isotherm for the non-Gaussian engine with $\kappa = 10$ for the hot reservoir in Supplementary Fig. 4d. Once again, the violation of quasistaticity (gray-shaded region) is concurrent with the onset of irreversibility.

**Tuning the performance of a Stirling engine through memoryless non-Gaussian noise.** We now examined how differences in the nature of noise statistics affected the power output of our engines. In the quasistatic limit $P(\tau) = -\frac{\langle W_{\text{cyc}}(\tau) \rangle}{\tau} \rightarrow 0$ since $\tau \rightarrow \infty$, while at high cycle frequencies $W_{\text{diss}}$ is large and $P$ is once again small. At intermediate $\tau$, however, these effects compete resulting in a maximum in $P$ and this is a feature of both macroscopic and mesoscopic engines[2,20]. Figure 3a shows the most

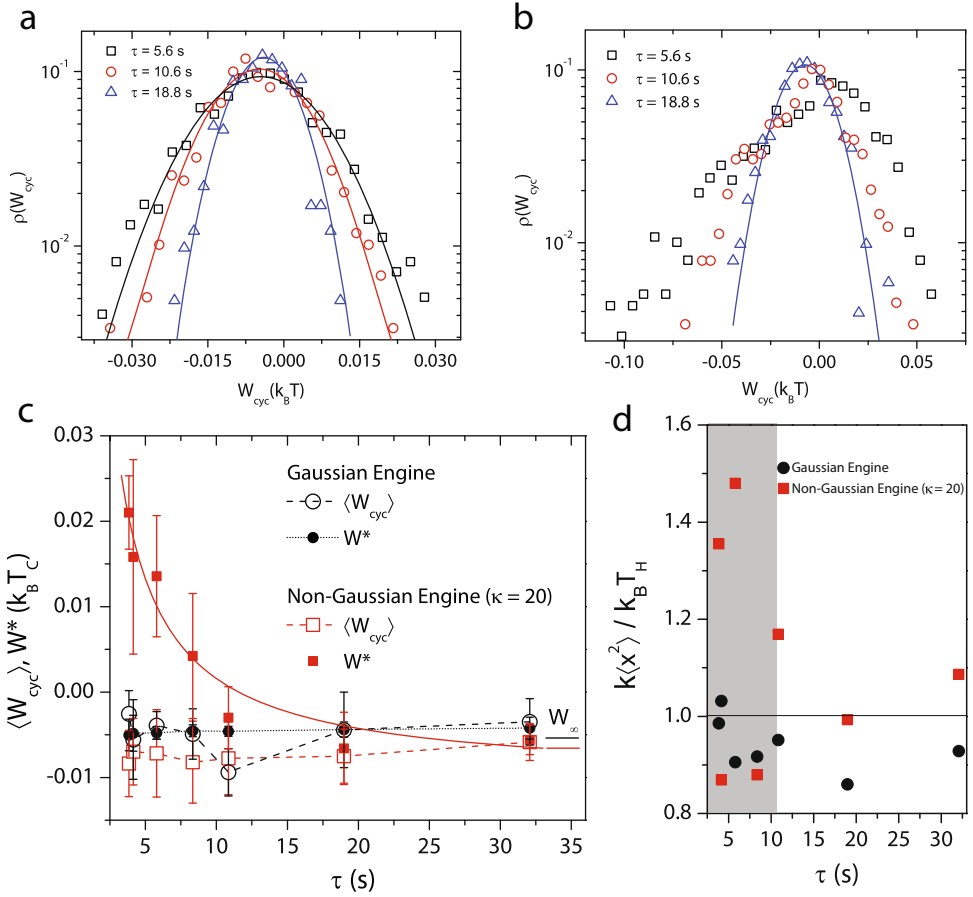

**Fig. 2 Buildup of irreversibility in the non-Gaussian Stirling engine at finite $\tau$.** In **a** and **b**, we show probability distribution of work done per cycle $\rho(W_{cyc})$ for the Gaussian engine and for the non-Gaussian engine with $\kappa = 20$ in the hot reservoir, respectively, for different cycle durations. $\tau = 18.8$ s (blue triangles), $\tau = 10.6$ s (red circles), and $\tau = 5.6$ s (black squares). Solid lines represent corresponding Gaussian fits to the data. **c** Red hollow and solid squares show the average work done per cycle $\langle W_{cyc} \rangle$ and the most-probable work $W^*$, respectively, for the non-Gaussian engine with $\kappa = 20$ for the hot reservoir at various $\tau$. The red solid line is a fit to Eq. (1). Black hollow and solid circles show $\langle W_{cyc} \rangle$ and $W^*$ respectively for the thermal/Gaussian engine. At large $\tau$, the experimentally calculated work for these engines agrees with theoretically calculated quasistatic work $W_\infty$ indicated by the small red horizontal line for the non-Gaussian engine with $\kappa = 20$ for the hot reservoir and by the black line for the Gaussian engine. Mean work $\langle W_{cyc} \rangle$ is calculated for each realization of the engine over 450 cycles for $\tau = 3.7$ s, 400 cycles for $\tau = 4$ s, 278 cycles for $\tau = 5.6$ s, 193 cycles for $\tau = 8$ s, 150 cycles for $\tau = 10.6$ s, 85 cycles for $\tau = 18.8$ s and 50 cycles for $\tau = 32$ s. **d** The ratio $k\langle x^2 \rangle / k_B T_{eff}^H$ calculated at the midpoint of the hot isotherm for various $\tau$ is showed by the red squares for the non-Gaussian engine with $\kappa = 20$ in the hot reservoir and by the black circles for the Gaussian engine. The horizontal line indicates the equilibrium condition, which is violated inside the shaded gray region, in the case of the non-Gaussian engine with $\kappa = 20$ in the hot reservoir. The error bars indicate the standard deviations of the mean and the most probable quantities across different experiments.

probable power, $P^*(\tau) = \frac{-W^*(\tau)}{\tau}$, for the Gaussian Stirling engine (circles) and for the non-Gaussian one with $\kappa = 10$ (triangles) and $\kappa = 20$ (squares) for the hot reservoir, respectively. Since for the Gaussian engine, over the range of $\tau$ studied $\Sigma = 0$, $P^*(\tau)$ is same as $P(\tau)$ and only increases monotonically on lowering $\tau$. Whereas for the non-Gaussian engine on reducing $\tau$, $P^*(\tau)$, especially for the engine with $\kappa = 10$ for the hot reservoir, first increases and crosses zero for $\tau \approx 6$ s indicating stalling of the engine. Although we do not evidence a clear maximum for the non-Gaussian engine with $\kappa = 20$ for the hot reservoir, $P^*(\tau)$ becomes negative for $\tau < 10$ s. We note that the primary contribution to irreversibility stems from the inability of the particle to explore the available volume during the isothermal expansion step. Better volume equilibration can be achieved by operating the engine across baths at higher temperatures. Our Gaussian engine however operates across baths at effective temperatures lower than the non-Gaussian one. Thus, the maximum in $P$ for a hypothetical Stirling engine operating across Gaussian baths with effective temperatures identical to either of the two non-Gaussian ones

should be at a $\tau$ that is smaller than the one for the Gaussian engine studied here. However, even for the smallest cycle duration investigated here, we did not evidence a maximum in $P$ for the Gaussian engine (circles in Fig. 3a). Thus, even without memory, altering the statistical properties of the noise bath alone allows for tuning the performance characteristics of mesoscopic heat engines.

For a complete understanding of the operation of the non-Gaussian engines, we calculated their efficiency at various $\tau$ and benchmarked it with the thermal engine. Conventionally, the efficiency, $\varepsilon = \frac{W_{cyc}}{Q}$, where $Q$ is the heat absorbed by the particle when it is in contact with the hot reservoir. $Q$ is the sum of the isochoric heat during the transition from state point ②–③, given by

$$Q_{2 \to 3} = -\frac{1}{2} k_{max} (T_{eff}^H - T_{eff}^C) \qquad (2)$$

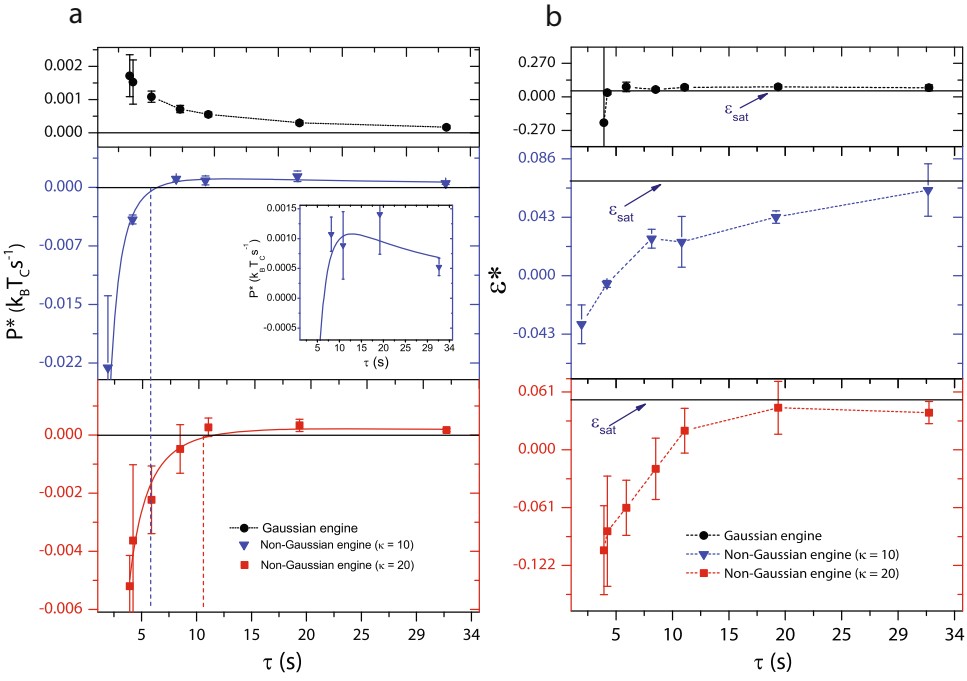

**Fig. 3 Quantifying the performance of the non-Gaussian Stirling engine. a** The most probable power $P^*$ with $\tau$. Black circles represent $P^*$ for the Gaussian engine, blue triangles represent $P^*$ for the non-Gaussian engine with $\kappa = 10$ at the hot reservoir and red squares represent $P^*$ for the non-Gaussian engine with $\kappa = 20$ at the hot reservoir. With decreasing $\tau$, $P^*$ increases considerably (inset) for the non-Gaussian engine with $\kappa = 10$ at the hot reservoir and rapidly falls off for $\tau \leq 8$ s. For the non-Gaussian engine with $\kappa = 20$ at the hot reservoir, the increase in $P^*$ is rather small and it decays for $\tau \leq 10.6$ s. The blue and red solid lines are calculated from the fit to Eq. (1) and are overlaid on the experimental data. The blue (red) vertical dashed line indicates the $\tau$ below which $P^*$ is negative for the non-Gaussian engine with $\kappa = 10$ ($\kappa = 20$) at the hot reservoir. **b** The most-probable efficiency $\varepsilon^*$ for various $\tau$. Black circles represent $\varepsilon^*$ for the Gaussian engine, blue triangles represent $\varepsilon^*$ for the non-Gaussian engine with $\kappa = 10$ at the hot reservoir and red squares represent $\varepsilon^*$ for the non-Gaussian engine with $\kappa = 20$ at the hot reservoir. The blue solid lines indicate the theoretically calculated Stirling saturation, $\varepsilon_{Sat}$. Efficiency $\varepsilon_{Max}$ just before the rapid drop in power at $\tau = 8$ s ($\tau = 10.6$ s) of the non-Gaussian engine with $\kappa = 10$ ($\kappa = 20$) at the hot reservoir agrees with the Curzon–Ahlborn efficiency $\varepsilon_{CA}$. The error bars indicate the standard deviations of the mean and the most probable quantities across different experiments. Note that the black vertical line through the first data point (smallest $\tau$) is a portion of a large error bar. The error bars at other $\tau$ values are smaller than the symbol size.

and the isothermal heat during transition from ③–④, given by

$$Q_{3 \to 4} = \int_{(3)}^{(4)} \frac{\partial U}{\partial x} \dot{x} dt = W_H + Q_{boundary} \qquad (3)$$

Here, $W_H = \frac{1}{2} \int_{(3)}^{(4)} x^2 \circ dk$ is the work done in the hot isotherm and $Q_{boundary} = -\frac{1}{2}[k(t)x^2(t)]_{(3)}^{(4)}$. For the non-Gaussian engine, we naturally chose $W^*$ instead of $W_{cyc}$ and defined the most probable efficiency $\varepsilon^* = \frac{W^*}{\langle W_H \rangle + \langle Q_{boundary} \rangle + \langle Q_{isochoric} \rangle}$ (see Supplementary Note 5). For the Gaussian engine, the experimentally determined $\varepsilon^*$ (black circles in Fig. 3b) hovers around the theoretically calculated saturation Stirling efficiency $\varepsilon_{Sat} = \varepsilon_c [1 + \frac{\varepsilon_c}{\ln (k_{max}/k_{min})}]^{-1}$ (solid blue line). Here, $\varepsilon_c = 1 - \frac{T_{eff}^C}{T_{eff}^H}$ is the Carnot efficiency. Whereas for the non-Gaussian engines with $\kappa$ of 10 and 20 for the hot reservoirs, $\varepsilon^*(\tau)$ converges to $\varepsilon_{Sat}$ only at large $\tau$ (blue triangles and red squares in Fig. 3b). When $\tau$ is reduced, $\varepsilon^*(\tau)$ drops and becomes negative for $\tau < 6$ s for the non-Gaussian engine with $\kappa = 10$ and $\tau < 10$ s for the non-Gaussian one with $\kappa = 20$ indicating stalling of the engines. Of particular importance in the operation of real heat, engines are the efficiency at maximum power $\varepsilon_{Max}$. Most remarkably, for both the non-Gaussian engines the experimentally determined efficiency values agree within error bars with the theoretically predicted Curzon–Ahlborn efficiency, $\varepsilon_{CA} = \frac{\varepsilon_{Sat}}{2 - \alpha \varepsilon_{Sat}} = 0.035$ ($\kappa = 10$) and 0.026 ($\kappa = 20$)[19,20]. In our experiments, $\alpha$ is a constant calculated from the irreversibility parameters corresponding to the work

done in the hot and cold isotherms (Supplementary Fig. 5 and Supplementary Note 6). While it is known that $\varepsilon_{Max} \approx \varepsilon_{CA}$ for both macro and mesoscopic thermal engines, ours is the first observation of this being the case even for a non-Gaussian engine.

## Discussion

Collectively, our experiments show that a micrometer-sized Stirling engine operating between a Gaussian and a non-Gaussian bath, without memory, indeed performs like a conventional engine in the quasistatic limit as anticipated by theory. On lowering the cycle times, the buildup of irreversibility in the engine, due entirely to the non-Gaussian nature of noise, results in work distributions that become increasingly negatively skewed, unlike a thermal engine where it remains Gaussian. Strikingly, this noise-induced enhancement of irreversibility modulates the performance characteristics of the non-Gaussian engine in a manner similar to predictions by Curzon and Ahlborn for thermal engines where irreversibility sets in purely due to the rapid change of the control parameter. Our experiments thus reveal a strategy for optimizing the performance of a mesoscale engine by tuning only the nature of noise statistics. Importantly, the ease with which the noise can be engineered and also applied locally, i.e., on the particle scale, in our approach presents advantages over other reservoir engineering methods where this can prove to be difficult, if not impossible[6,21]. This should now make feasible the experimental realization of future stochastic machines like the non-Gaussian and the Buttiker–Landauer ratchet[22–24].

## Methods

**Experimental set-up for reservoir engineering.** In order to impart additional noise into the trapped colloid, a secondary optical trap was flashed along a line passing through the time-averaged center of the particle at variable distances from the same. This was achieved by coupling a second laser (Excelsior 1064 nm, Spectra Physics USA) to the microscope which is reflected from an SLM. The SLM contains a $512 \times 512$ array of shiny electrodes covered with a transparent liquid crystal layer so that an electric potential modulation across the electrodes imposes an additional phase pattern on the incident beam. We carried out two sets of experiments with two different SLMs. The experiments for the Gaussian engine and the non-Gaussian engine with $\kappa = 20$ for the hot reservoir were carried out with a Boulder Nonlinear Systems USA SLM which upon interfacing to a computer through MATLAB could flash a series of desired phase patterns at a fixed frequency of maximum 34 Hz. For the non-Gaussian engine with $\kappa = 10$ for the hot reservoir, we used an SLM from Meadowlark Optics USA to flash phase patterns at a fixed frequency of 135 Hz. The use of SLMs enabled us to dynamically reconfigure the position of the first-order diffraction spot by applying a series of linear diffraction grating patterns with a varying periodicity which is controlled through a computer. We blocked the zeroth-order spot so that only the first order spot is incident on the back of the microscope objective resulting in a flickering optical trap in the vicinity of the tweezed colloidal particle.

**Image acquisition and processing.** Images of the trapped colloid were captured at 250 Hz using a fast camera (Photron 500K-M3) attached to the microscope. The position of the particle's center in each frame was located at the subpixel level using the particle tracking codes by R. Parthasarathy[25]. This allowed us to find the particle's position with an accuracy of 5 nm.

**Non-Gaussian Reservoir Engineering.** For engineering the non-Gaussian reservoir, $\delta a$ were chosen from a $\delta$-correlated distribution with zero mean and skewness but a high $\kappa = 50$. One such distribution (for 34 Hz noise) with a standard deviation of $\sigma = 0.28$ μm for engineering a reservoir with $\kappa = 20$ is represented in Supplementary Fig. 2b. To create this distribution, we first generate two highly asymmetric distributions $\delta a_L$ and $\delta a_R$ (Supplementary Fig. 2a) with a standard deviation of 0.28 μm, a $\kappa = 60$ and skewness of $-6.5$ for $\delta a_L$ and $+6.5$ for $\delta a_R$ through Pearson's protocol in MATLAB. Next, we add/subtract a suitable number to $\delta a_L$ and $\delta a_R$ so that their peaks coincide at zero. Then we take the union of $\delta a_L$ and $\delta a_R$ and randomly permute all the elements to finally obtain the set of $\delta a$. In order to realize a desired effective temperature with such a noise, the standard deviation of $\delta a$ is optimized. As shown in Supplementary Fig. 2c, $\rho(\delta a)$ for the non-Gaussian reservoir with $\kappa = 10$ is similar to the one in Supplementary Fig. 2b except for the broad central portion. Since, for the case of Supplementary Fig. 2c, the flashing frequency is 135 Hz and the stiffness of the flashing trap is also higher, a similar-looking $\rho(\delta a)$ results in a different $\rho(x)$ with $\kappa = 10$. It should be noted that heavy tails rise due to rare events that can only be captured with huge statistics. Since we are limited by finite flashing frequencies, it is not possible to completely sample the statistics within one isotherm even for the largest $\tau$. To address this issue, the engine was cycled enough times (depending on $\tau$) so that the collection of all the hot isotherms exhausts all the rare events.

**Instantaneous isochoric transitions.** The isochoric transitions ②→③ and ④→① shown in Fig. 1e of the main text are realized by changing the statistics and the variance of $\delta a$-distribution. The transition ②→③ is realized by changing the $\delta a$ distribution from a Gaussian resulting in $T_{eff} = 1570$ K to a non-Gaussian producing $T_{eff} = 1824.3$ K while the transition ④→① is realized by the reverse. Since the secondary laser is diffracted by a computer-controlled SLM, the distribution from which $\delta as$ are chosen can be altered in 1/34th of a second. Thus the particle is decoupled and coupled from one engineered reservoir to the other in less than 33 ms which is almost negligible even in compared to the lowest cycle time and hence instantaneous.

## Data availability

The datasets generated during and/or analyzed during the current study are not publicly available due to the very large size of the video files (>600 Gb) but are available from the corresponding author on reasonable request.

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

## Acknowledgements

N.R. thanks Dr. Sudeesh Krishnamurthy for fruitful discussions. N.R. thanks Jawaharlal Nehru Center for Advanced Scientific Research (JNCASR) for financial support. A.K.S. thanks the Department of Science and Technology (DST), Government of India for a Year of Science Fellowship. R.G. thanks JNCASR for financial support.

## Author contributions

N.R., N.L., A.K.S. and R.G. designed the experiments. N.R., N.L. and R.G. devised the experimental procedures. N.R. performed the experiments and carried out the data analysis. N.R. and R.G. wrote the paper with inputs from N.L. and A.K.S.

## Competing interests

The authors declare no competing interests.
