## [Peer Review File · Nature Communications]

REVIEWER COMMENTS

Reviewer #1 (Remarks to the Author):

\begin{document}

This article describes an experiment in which a trapped Brownian bead is used to study the Stirling cycle working between two thermal baths characterized by either Gaussian or non-Gaussian thermal baths. The idea is interesting and new and its principle it might be published in NCOMMS, but I have several doubts about the experiment and the text that the authors have to clarify.

\begin{itemize}

\item[1] The first is about the flashing laser. I understand that 34Hz is imposed by the Spatial Light Modulator which is a slow device but this frequency is rather small and certainly not fully filtered by the particle. How long is the laser flashed at each pulse? My point is that because of the flashing and the rather small frequency the stiffness changes intermittently and there is a component of the noise which is multiplicative not only additive as the author says. This will change completely the theoretical description and I am not sure that the noise is fully uncorrelated. Why the spectra of the bead in supplementary material are plotted only till 20 Hz when the sampling frequency is much higher. The spectra till at least 50Hz has to be plotted to check whether there are anomalies, especially in the non-Gaussian case.

\item[2] Fig.2b. In the text it is said that at 32s for the non-Gaussian case the work fluctuations are still Gaussian. Looking at the figure with 8 points highly dispersed, it seems to me a too strong claim.

\item[3] Fig.2d. 'Strongly violated inside the shaded area' is another very strong claim. I see only three red points which violate the equilibrium conditions. The other are within the statistical errors.

\item[4] Eq.1 is not empirical but has a theoretical reason. See Sekimoto articles and book.

\item[5] Page 8, Fig.3a. 'Whereas the non-Gaussian engine, on reducing τ , $\langle P(\tau) \rangle$ first appears to increase slightly'. With the error bars in the figures and with only two points, it is not possible to write such a claim.

\item[6] Page 8. 'For the non Gaussian we naturally chose $\langle W \rangle$ instead of $\langle W_{cyc} \rangle$.' Why? where does this claim come from? As there is a highly fluctuating quantity the mean is more meaningful than the peak. What do the authors want to prove? This brings me to the abstract because using $\langle W_{cyc} \rangle$ the conclusions are probably not the same. Why the heat versus $\langle \tau \rangle$ is never plotted?

\item[7] In supplementary material Fig.4. In the caption it is written 'work' in the vertical axis is a power. What is this?

\item[8] In supplementary material. If $\langle W_H \rangle$ and $\langle W_C \rangle$ are fitted by eq. 6 and eq.7, how do the authors pretend to see a maximum in Fig.3a? (see my point 5)

\end{itemize}

\end{document}

Reviewer #2 (Remarks to the Author):

The authors investigate experimentally the consequences of non-Gaussian environments on the performance of a Stirling engine, where the working fluid is a trapped colloidal particle. To draw conclusions, they compare a Stirling engine involving only Gaussian environments (at two different temperatures for achieving the cycle) to an engine where the colloidal particle is in contact with a Gaussian and a non-Gaussian environments, corresponding to the cold and hot environments respectively.

In the context of stochastic thermodynamics and macro/mesoscopic engines, understanding the role of fluctuations on the performance of engines is presently a very active research direction. It is of fundamental importance to understand how fluctuations and more generally the statistics of environments in contact with nano/microscale machines influence their functioning. This very fundamental question is valid both in the classical and quantum regime, and this work provides key results for a better understanding of the role of statistics when manipulating single constituents in the classical regime. As such, this work is of interest for a wide range of researchers.

I have a number of questions - comments - suggestions that I list below in order of appearance. I would recommend publication of the manuscript with minor modifications. Let me mention that I am a theorist, and hence can not judge very profoundly the experimental techniques.

Here is my list.

1) I have some concern with the use of “equilibrium” by the authors. Similarly to other fundamental concepts in physics, it makes take a slightly different meaning from one community to another, and it is maybe the case here. In a mesoscopic physics sense, a reservoir at equilibrium is a reservoir with a well-defined temperature (and chemical potential). Depending on the nature of the reservoir itself (for instance bosonic or fermionic), one uses the adequate distribution. As the authors explain, their protocol to engineer non-Gaussian distribution for the particle’s position is compatible with assigning a well-defined effective temperature to it and having a delta-correlated correlation function for it (no memory). The protocol only affect the statistics, and therefore I would tend to think that it can be considered as a non-Gaussian reservoir AT equilibrium.

With respect to one of the sentence in the abstract “[...] with noise statistics that is non-Gaussian [...], the simplest departure from equilibrium, [...]”, I would also be careful. What I would have in mind to probe an out-of-equilibrium situation just from the statistics would be to consider two reservoirs at the same temperature, but with different statistics (for instance Gaussian and non-Gaussian) and see whether they still induce some transport or non-equilibrium physics through the working fluid.

Could the authors comment on this general comment? If it’s a matter of “community-dependent” langage, then it would be good to clarify it and to avoid using it right in the abstract. If I am misleading, please comment. If it was some abuse, please correct.

2) p.3, l.4: work and efficiency are not parameters!

3) Active vs. Passive: not introduced in the main text. Although the reader guesses what it refers to, it would

be preferable to clearly define what you mean.

With respect to that, one sentence may be confusing, see p.2, before ref [15]. There, you say that memory is a typical feature of active baths. And in your work, you emphasise a lot the memoryless aspect of the environments while having a non-Gaussian statistics and this is what you refer to “active ” in Fig. 2 and Fig. 3 for instance.

4) Fig. 1 Panels d) and e): for which cycle duration τ do the data correspond?

Average over how many cycles?

5) p.7: description of fig. 2d: inverted squares and circles for non-Gaussian and Gaussian.

6) p.8 : Discussion of Fig. 3 a: the dependance of $P^*(\tau)$ on τ is one of the most important results. However, on the figure, the increase and then decrease can be barely seen. Whereas the increase /decrease is very clear at low τ , the non-monotonic behaviour is not convincing. Would there be another way to evidence it? Maybe by showing the derivative of P^* as a function of τ ? This could be a way to better quantify when non-Gaussianity plays a role and could be use for instance in Fig 2 d to determine the grey area?

9) Related question to point 8) above: I was wondering if it would be possible to quantify why the non-Gaussian features do clearly show up for $\tau \sim 10$ s. This value is probably related to the parameters of the experiment, especially compared to the other time scales of the system. Could the authors comment? Is this specific τ associated to the specific profile of δa ?

10) Also related comment:

What would be the protocol to achieve another characteristic cycle time for the efficiency at max power? Do the authors have data?

I would find this very interesting as it would make the claim that you can control the performances of the machine through reservoir engineering even stronger.

11) General comment: the feature “memoryless” of the experiment is emphasised a lot in the title and by using italic font. I understand from the intro why this is a key feature for demonstrating the novelty of these results, however I feel it is not appropriately justified in the main text. While being discussed in Supp. Note 1, it could be helpful to explain it in a concise way in the main text.

12) In the Methods about the reservoir engineering, they mention the importance of the number of cycles N to average over as a function of τ . I believe this is an important information that should be present at least in the captions of the different figures for the various τ .

13) Fig. 2d: I find the data (circle) that should satisfy equilibrium not so convincing. Could the authors maybe comment or explain the large deviation wrt 1, especially for long cycles?

14) Fig. 4: Do Panels a and b also show the data from Fig 2a and 2b, i.e. for cycles with $\tau = 5.6, 10.6$ & 32 ? Does not look so. Is there a reason? And also again my remark about passive vs. active.

Typos:

- p.2 ... where the displacement ... is a Gaussian, -> ... where the displacement ... is Gaussian,
- Several places in the text: "sans memory" -> without memory?
- p.6: before subsection "elucidating ...": were near instantaneous" -> were nearly instantaneous
- p.8: ... which is the same as same as ... -> which is the same as ...
- Author contribution: from all N.L. and A.K.S -> from N.L. and A.K.S

Response to REVIEWER'S COMMENTS

Reviewer #1:

Comment: *This article describes an experiment in which a trapped Brownian bead is used to study the Stirling cycle working between two thermal baths characterized by either Gaussian or non-Gaussian thermal baths. The idea is interesting and new and in principle it might be published in NCOMMS, but I have several doubts about the experiment and the text that the authors have to clarify.*

Response: We thank the reviewer for finding our study interesting, new and that in principle it can be published in Nature Communications. Following the reviewer's comments and those from Reviewer #2, we have now done a completely new set of experiments that reinforces our original findings. A detailed point-by-point response to this reviewer's comments is provided below.

Comment 1: *The first is about the flashing laser. I understand that 34Hz is imposed by the Spatial Light Modulator which is a slow device but this frequency is rather small and certainly not fully filtered by the particle. How long is the laser flashed at each pulse? My point is that because of the flashing and the rather small frequency the stiffness changes intermittently and there is a component of the noise which is multiplicative not only additive as the author say. This will change completely the theoretical description and I am not sure that the noise is fully uncorrelated. Why the spectra of the bead in supplementary material are plotted only till 20 Hz when the sampling frequency is much higher. The spectra till at least 50Hz has to be plotted to check whether there are anomalies, especially in the non-Gaussian case..*

Response 1: The reviewer is indeed correct that the 34 Hz noise imposed by the SLM is slow in comparison to many other devices used to manipulate laser beams. When we started these experiments in 2018 this was the only device available to us. More recently, we have acquired a faster SLM and we have carried out a fully new set of experiments with the newer SLM with a noise frequency of 135 Hz.

With the older SLM the pulse duration for each flash is 33 ms. We would however like to point out that the PSD of the particle for both Gaussian and non-Gaussian noise shows deviations from a Lorentzian beyond 20 Hz (see Figure below) and hence the deviation is not dependent on the nature of the noise. Also, we see the deviations only after about an order-of-magnitude fall in the PSD (**Supplementary Fig. 3a**). We also like to point out here, that for both the Gaussian and non-Gaussian engines, the work and efficiency for large cycle durations determined from the experimental data is in excellent agreement with the theoretically predicted quasistatic Stirling output (**Figure 2c of the Main manuscript**). Since the trap stiffness appears in the theoretical expression, the agreement between theory and experiment suggests that at least for the cycle durations studied, the noise can be assumed to be additive. It is possible that the effects suggested by the referee may kick in at much smaller cycle durations.

To resolve this issue nonetheless, **we have carried out a total of 20 new experiments for 6 different Stirling cycle durations with a newer and faster SLM with a flash frequency of 135 Hz**. For this SLM the PSD (see Figure below) decays by more than two-orders of magnitude before showing deviations from the Lorentzian beyond 70 Hz (**new Supplementary Fig. 3b**) of the revised manuscript. Using the newer SLM, we now operated a Stirling engine operating between a non-Gaussian reservoir with kurtosis, $\kappa = 10$, and a Gaussian reservoir. The kurtosis of the hot reservoir for the previous non-Gaussian engine was 20. Once again, in the quasistatic limit the experimentally determined work and efficiency are in excellent agreement with theoretical predictions (**new Supplementary Fig. 4**). This further strengthens the fact that the noise for reservoir engineering is

indeed like a thermal noise which is uncorrelated and additive. **The necessary changes made to the main text and supplement are now shown in blue color.**

Figure: (a) and (b) shows the PSD of a trapped colloidal particle for a flash frequency of 34 Hz and 135 Hz, respectively. Blue lines show Lorentzian fits to the data. The agreement with the fit is over about 1-decade for 34 Hz noise and 2-decades for the 135 Hz noise.

Comment 2: Fig.2b. In the text is said that at 32s for the non-Gaussian case the work fluctuations are still Gaussian. Looking at the figure with 8 points highly dispersed, it seems to me a too strong claim.

Response 2: We agree with the reviewer that the work done for 32 s cycle duration lacks statistics for a functional fit. For the largest cycle durations, our experiments are data consuming and limited by the on-board camera storage capacity. In the **Revised Manuscript and Supplement**, we show the Gaussian fits of $p(W_{cyc})$ for 18.8 s cycle duration for both the Gaussian and the non-Gaussian engines (**new Fig. 2b and Supplementary Fig. 4b**). For all these engines the work distributions are Gaussian. This shows that in the quasistatic limit, where irreversibility is negligible, even for the non-Gaussian engine work distribution is a Gaussian.

Comment 3: Fig.2d. 'Strongly violated inside the shaded area' is another very strong claim. I see only three red points which violate the equilibrium conditions. The other are within the statistical errors.

Response 3: We agree with the reviewer and we have removed the word 'strongly'. We only wished to point out here that when the most-probable work W^* becomes positive for the non-Gaussian engine, this coincides with the violation of equilibrium (points in the grey-shaded region of Fig. 2d). To strengthen this observation, in the **new Supplementary Fig. 4d**, we plot $k \langle x^2 \rangle / k_B T_H$ for the non-Gaussian engine built using the faster SLM. Here as well $k \langle x^2 \rangle / k_B T_H$ deviates largely from 1 below a cycle duration of 8s, which is again consistent with the lift-off of W^* .

Comment 4: Eq.1 is not empirical but has a theoretical reason. See Sekimoto articles and book.

Response 4: We apologize for the oversight. We have removed the word "empirical" from the main text (Line 136 of Revised Manuscript).

Comment 5: Page 8, Fig.3a. 'Whereas the non-Gaussian engine, on reducing τ , $\$P(\tau)\$ first appears to increase slightly'. With the error bars in the figures and with only two points, it is not possible to write such a claim.$

Response 5: We would like to emphasize that the absolute value of the power maximum depends on the experimental parameters. We agree to the reviewer that the power peak is not prominent for the non-Gaussian engine with $\kappa = 20$ for the hot reservoir. We have now rewritten this sentence (**Line 189-191 of Revision**). However, for the new set of non-Gaussian engine experiments with κ

= 10 for the hot reservoir, the maximum in power is rather enhanced and this is shown in the inset of **new Figure 3a**.

Comment 6: Page 8. 'For the non Gaussian we naturally chose $\langle W^* \rangle$ instead of $\langle W_{cyc} \rangle$.' Why? where does this claim come from? As there is an highly fluctuating quantity the mean is more meaningful than the peak. What do the author want to prove?. This bring me to the abstract because using $\langle W_{cyc} \rangle$ the conclusions are probably not the same. Why the heat versus τ is never plotted?

Response 6: We find that $\langle W_{cyc} \rangle$ for not just the Gaussian engine but also for both the non-Gaussian engines remains more or less constant on reducing cycle duration. This is clearly not consistent with the violation of quasistaticity at small cycle durations for the non-Gaussian engines (see **Fig. 2d** and **new Supplementary Fig. 4d**). On the other hand, the most-probable work, W^* , in fact becomes positive in the regime of cycle durations where we see equilibrium violation. This shows that W^* is a more meaningful quantity to use for non-Gaussian engines. For a Gaussian/thermal engine, this is never the case no matter how small the cycle duration becomes and $\langle W_{cyc} \rangle$ and W^* are identical. This is also consistent with the intuition that unless the rarest of rare fluctuations are sampled, which is not possible for small cycle durations unless an infinite number of cycles are performed, W^* is a more useful metric of engine performance. This according to us is a feature unique to non-Gaussian engines.

The heat in the hot isotherm is evaluated as mentioned in Supplementary Note 5. For all the engines performed, $Q_{isochoric}$ and $Q_{boundary}$ has a symmetric distribution with mean at $-k_B(T_{H\text{eff}} - T_{C\text{eff}})$ and zero, respectively. Therefore, the only contribution to the heat that depends upon the cycle duration and the statistics is the work done in the hot-isotherm, W_H . We had indeed plotted the most-probable value W_H^* as a function of cycle duration in **Supplementary Fig. 4a of the previous submission**. In the Revised Supplement, we show W_H^* for both the non-Gaussian engine with kurtosis of 10 and 20 at the hot reservoir in **new Supplementary Fig. 5**. We also show the work done in the cold-isotherm, W_C^* , for both the engine as this is utilized in the calculation of Curzon-Alborn efficiency.

Comment 7: In supplementary material Fig.4. In the caption is written 'work' in the vertical axis is a power. What is this?

Response 7: We apologize for the typo where we accidentally used units of power. The vertical axes in Supplementary Fig. 4 of the first version of the manuscript represents the most-probable work that has been performed along a single isotherm. In the revised version of the manuscript, these plots of most-probable work per isotherm have been **moved to Supplementary Fig. 5c and 5d**. Whereas, **Supplementary Fig. 5a and 5b** show the most-probable work in the hot and cold isotherms of the non-Gaussian engine with $\kappa=10$ for the hot reservoir, respectively.

Comment 8: In supplementary material. If $\langle W_H \rangle$ and $\langle W_C \rangle$ are fitted by eq. 6 and eq.7, how the authors pretend to see a maximum in Fig.3a? (see my point 5)

Response 8: As mentioned in response to Comment 7, the vertical axis in Supplementary Fig. 4 of the first version of the manuscript (**Supplementary Fig. 5 of Revision**), is the most-probable work during the isothermal expansion and compression in the hot and the cold reservoir, respectively, and not the power. The fits shown in this figure are according to equation (6) and (7) of the Supplementary Information. Whereas Fig 3a plots the most-probable power calculated as the W^*/τ . The fit shown in Fig. 3a of the main text is according to equation (1) of the main text divided by the cycle duration.

We hope the reviewer is satisfied with our responses to their comments and the manuscript is acceptable for publication in Nature Communications.

Reviewer #2

Comment: *The authors investigate experimentally the consequences of non-Gaussian environments on the performance of a Stirling engine, where the working fluid is a trapped colloidal particle. To draw conclusions, they compare a Stirling engine involving only Gaussian environments (at two different temperatures for achieving the cycle) to an engine where the colloidal particle is in contact with a Gaussian and a non-Gaussian environments, corresponding to the cold and hot environments respectively.*

In the context of stochastic thermodynamics and macro/mesoscopic engines, understanding the role of fluctuations on the performance of engines is presently a very active research direction. It is of fundamental importance to understand how fluctuations and more generally the statistics of environments in contact with nano/microscale machines influence their functioning. This very fundamental question is valid both in the classical and quantum regime, and this work provides key results for a better understanding of the role of statistics when manipulating single constituents in the classical regime. As such, this work is of interest for a wide range of researchers.

I have a number of questions - comments - suggestions that I list below in order of appearance. I would recommend publication of the manuscript with minor modifications. Let me mention that I am a theorist, and hence can not judge very profoundly the experimental techniques.

Response: We thank the reviewer for finding our work to be of interest and recommending publication of the manuscript in Nature Communications. We provide below a point-by-point response to their comments.

Comment 1: *I have some concern with the use of “equilibrium” by the authors. Similarly to other fundamental concepts in physics, it makes take a slightly different meaning from one community to another, and it is maybe the case here. In a mesoscopic physics sense, a reservoir at equilibrium is a reservoir with a well-defined temperature (and chemical potential) Depending on the nature of the reservoir itself (for instance bosonic or fermionic), one uses the adequate distribution. As the authors explain, their protocol to engineer non-Gaussian distribution for the particle’s position is compatible with assigning a well-defined effective temperature to it and having a delta-correlated correlation function for it (no memory). The protocol only affect the statistics, and therefore I would tend to think that it can be considered as a non-Gaussian reservoir AT equilibrium.*

With respect to one of the sentence in the abstract “[...] with noise statistics that is non-Gaussian [...], the simplest departure from equilibrium, [...]”, I would also be careful. What I would have in mind to probe an out-of-equilibrium situation just from the statistics would be to consider two reservoirs at the same temperature, but with different statistics (for instance Gaussian and non-Gaussian) and see whether they still induce some transport or non-equilibrium physics through the working fluid.

Could the authors comment on this general comment? If it’s a matter of “community-dependent” langage, then it would be good to clarify it and to avoid using it right in the abstract. If I am misleading, please comment. If it was some abuse, please correct.

Response 1: We thank the reviewer for this comment. There might have been some abuse of usage in our first submission. We do agree with the reviewer that any bath with statistics resulting in well-defined effective temperature is essentially at equilibrium. Hence to remove ambiguity, we have

replaced the word “equilibrium” with “thermal/Gaussian” in all the necessary places in the **Revised Abstract and Main Text**. The changes made to the main text are shown in blue.

If an engine operates between two baths with well-defined effective temperatures (no matter the statistics), it has been proven theoretically (**Ref. [15] of the Main Manuscript**) that the average work done in the quasistatic limit equals its thermal counterpart. From this point of view, to extract energy on an average out of purely statistics seems impossible at least in the quasistatic limit. Based on our findings for the non-Gaussian engine at finite operating speeds, we anticipate that if not in an average sense, work in the most-probable sense can be extracted if the kurtosis of the “hot” reservoir is larger than that of the “cold” reservoirs.

Comment 2: *p.3, l.4: work and efficiency are not parameters!*

Response 2: We apologize for the error. This has now been corrected (**Line No: 4 of Revision**).

Comment 3: *Active vs. Passive: not introduced in the main text. Although the reader guesses what it refers to, it would be preferable to clearly define what you mean.*

With respect to that, one sentence may be confusing, see p.2, before ref [15]. There, you say that memory is a typical feature of active baths. And in your work, you emphasise a lot the memoryless aspect of the environments while having a non-Gaussian statistics and this is what you refer to “active” in Fig. 2 and Fig. 3 for instance.

Response 3: We thank the reviewer for this comment. We have now corrected this in our Revision, where we use “Gaussian” and “non-Gaussian” at the necessary places instead of “passive” and “active” respectively to avoid causing confusion.

Comment 4: *Fig. 1 Panels d) and e): for which cycle duration tau do the data correspond? Average over how many cycles?*

Response 4: In Fig. 1d, $\rho(x)$ is shown for the particle captured for a fixed stiffness of the central trap with noise from flashing traps and here no thermodynamic cycles are performed.

$\rho(x)$ of the particle as shown in Fig. 1e are the equilibrium distributions for the corresponding state-points of the Stirling cycle. These data were captured over 1500 s by keeping the stiffness of the confining potential fixed at respective values of the state points and changing the properties of the engineered reservoirs as needed for the specific state point. No Stirling cycles are involved in capturing these data and hence they reflect the true properties of the bath.

Comment 5: *p.7: description of fig. 2d: inverted squares and circles for non-Gaussian and Gaussian.*

Response 5: We apologize for the typo. This has now been corrected in the revision.

Comment 6: *p.8 : Discussion of Fig. 3 a: the dependance of $P^*(\tau)$ on τ is one of the most important results. However, on the figure, the increase and then decrease can be barely seen. Whereas the increase /decrease is very clear at low τ , the non-monotonic behaviour is not convincing. Would there be another way to evidence it? Maybe by showing the derivative of P^* as a function of τ ? This could be a way to better quantify when non-Gaussianity plays a role and could be use for instance in Fig 2 d to determine the grey area?*

Response 6: We agree with the reviewer that the non-monotonicity of P^* vs τ is not pronounced in Figure 3a of the main text in the first version of the manuscript. The non-Gaussian reservoir in this engine as presented in Figure 1e of the first version of the manuscript had a kurtosis of 20 for the hot reservoir. We would also like to emphasize that the absolute value of the peak of the power also

depends on the experimental parameters. Therefore, a mild peak could be a genuine feature of an engine. Taking the derivative with limited data points that are inherently noisy does not help in enhancing the effect.

In the revised version of the manuscript, we have presented a new set of experimental findings for a Stirling engine working between a hot non-Gaussian reservoir at 1500 K with a kurtosis, $\kappa = 10$, and a cold Gaussian reservoir at 1100 K (**new Fig. 3, new Supplementary Fig. 4 and Supplementary Note 4 of the Revised Manuscript**). This engine was operated with similar stiffness ratio and effective temperature difference as the previous non-Gaussian one. A higher speed Spatial Light Modulator (SLM) was used to impose noise at 135 Hz. The **inset to Figure 3a** of the revised version of the manuscript shows that the power peak in this case is pronounced and the non-monotonicity is more evident.

Comment 7: Related question to point 8) above: I was wondering if it would be possible to quantify why the non-Gaussian features do clearly show up for $\tau \sim 10$ s. This value is probably related to the parameters of the experiment, especially compared to the other time scales of the system. Could the authors comment? Is this specific τ associated to the specific profile of δa ?

Response 7: In the new set of experiments with kurtosis = 10 for the hot reservoir, we see that the crossover of W^* to a positive value happens at comparatively smaller cycle duration of $\tau = 6$ s (**new Supplementary Fig. 4c**). Correspondingly, the violation of equilibrium also starts for $\tau \leq 8$ s. The ratio of trap stiffnesses for both the non-Gaussian engines are identical. We would have anticipated that since the non-Gaussian engine with $\kappa = 10$ has a T_{eff}^H that is smaller than the non-Gaussian engine with $\kappa = 20$, the irreversibility for the former should set in before the latter. A higher temperature helps in volume equilibration and reduces irreversibility. This is clearly not the case and the cycle duration at which we see a peak in P^* is smaller for the non-Gaussian engine with $\kappa = 10$ in comparison to the one with $\kappa = 20$. This illuminates the crucial role played by the kurtosis of the distribution in deciding irreversibility and hence engine performance. A smaller kurtosis means a broader central portion of the PDF of the particle's position and results in more useful most-probable work at a small τ . Although the trend between the kurtosis and the non-Gaussian effects are evident, a precise relationship would be difficult to infer at this stage.

Comment 8: Also related comment:

What would be the protocol to achieve another characteristic cycle time for the efficiency at max power? Do the authors have data?

I would find this very interesting as it would make the claim that you can control the performances of the machine through reservoir engineering even stronger.

Response 8: We thank the reviewer for this comment. As mentioned in our responses to Comments 6 and 7 of this reviewer, we have performed a total of 20 new experiments for various cycle durations, the results of which are shown in (**new Fig. 3 and Supplementary Fig. 3, 4 and 5**). The power and efficiency of the three engines studied is plotted against cycle duration in **Figure 3** of the revised main manuscript. We find that the location of the peak power output of the non-Gaussian engine can indeed be tuned and these experiments establishes that the nature of noise - statistics is a parameter for tuning engine performance.

Comment 9: General comment: the feature "memoryless" of the experiment is emphasised a lot in the title and by using italic font. I understand from the intro why this is a key feature for demonstrating the

novelty of these results, however I feel it is not appropriately justified in the main text. While being discussed in Supp. Note 1, it could be helpful to explain it in a concise way in the main text.

Response 9: We thank the reviewer for the comment. We have now very concisely explained “memoryless” in **Line No 28-30 of the Main Manuscript**.

Comment 10: *In the Methods about the reservoir engineering, they mention the importance of the number of cycles N to average over as a function of τ . I believe this is an important information that should be present at least in the captions of the different figures for the various τ .*

Response 10: We thank the reviewer for this comment. The number of cycles for each of the cycle durations studies is mentioned in the **Caption to Fig. 2 of the Revised Manuscript**.

Comment 11: *Fig. 2d: I find the data (circle) that should satisfy equilibrium not so convincing. Could the authors maybe comment or explain the large deviation wrt 1, especially for long cycles?*

Response 11: The data presented in Fig. 2d was analysed over 50 cycles for all cycle durations to make comparisons possible. Although we perform ~ 500 cycles for the smaller cycle durations, at larger cycle durations, we are limited by the storage capacity of the camera. The fluctuations we see for longer cycles is due to poorer statistics. We would however like to point out that the violation from equilibrium at small cycle durations, which is our key finding, is clear for both the non-Gaussian engines (**Fig. 2d and new Supplementary Fig. 4d**)

Comment 12: *Fig. 4: Do Panels a and b also show the data from Fig 2a and 2b, i.e. for cycles with $\tau = 5.6, 10.6$ & 32 ? Does not look so. Is there a reason? And also again my remark about passive vs. active.*

Response 12: We do not have a Fig. 4 in our manuscript, and we guess the referee is alluding to Fig. 3. Data in all the figures are consistent and are shown for all the cycle durations studied. We apologize for the typo in this figure. It has now been changed to Gaussian and non-Gaussian as opposed to passive and active.

Typos:

- p.2 ... where the displacement ... is a Gaussian, -> ... where the displacement ... is Gaussian,

- Several places in the text: “sans memory” -> without memory?

- p.6: before subsection “elucidating ...”: were near instantaneous” -> were nearly instantaneous

- p.8: ... which is the same as same as ... -> which is the same as ...

- Author contribution: from all N.L. and A.K.S -> from N.L. and A.K.S

Response: We thank the referee for their careful reading of the manuscript. The typos have been corrected in the Revision.

We hope the reviewer is satisfied with our responses to their comments and the manuscript is acceptable for publication in Nature Communications.

REVIEWERS' COMMENTS

Reviewer #1 (Remarks to the Author):

The authors took seriously into account my comments and they answered in a rather convincing way presenting new results. The results of the article are reasonable and they might be useful for studying efficiency in complex environments. Thus they might have a general interest and the article can be published in the present form.

Reviewer #2 (Remarks to the Author):

The authors have carefully answered to all questions raised by the two referees. To my opinion, the manuscript has gain in precision and clarity. I would now recommend it for publication in Nature Communications. I would like to thank the authors for their precise answers with details and new data.

Response to Reviewers' comments:

Reviewer #1 (Remarks to the Author):

Comment: *The authors took seriously into account my comments and they answered in a rather convincing way presenting new results. The results of the article are reasonable and they might be useful for studying efficiency in complex environments. Thus they might have a general interest and the article can be published in the present form.*

Response: We are happy that the reviewer was convinced with our responses and thank them for recommending our manuscript for publication in Nature Communications.

Reviewer #2 (Remarks to the Author):

Comment: *The authors have carefully answered to all questions raised by the two referees. To my opinion, the manuscript has gain in precision and clarity. I would now recommend it for publication in Nature Communications. I would like to thank the authors for their precise answers with details and new data.*

Response: We thank the reviewer for the positive assessment of our previous response and for recommending the manuscript for publication.